# Molecular Detection and Genotyping of *Coxiella-*Like Endosymbionts in Ticks Collected from Animals and Vegetation in Zambia

**DOI:** 10.3390/pathogens10060779

**Published:** 2021-06-21

**Authors:** Toshiya Kobayashi, Elisha Chatanga, Yongjin Qiu, Martin Simuunza, Masahiro Kajihara, Bernard Mudenda Hang’ombe, Yoshiki Eto, Ngonda Saasa, Akina Mori-Kajihara, Edgar Simulundu, Ayato Takada, Hirofumi Sawa, Ken Katakura, Nariaki Nonaka, Ryo Nakao

**Affiliations:** 1Laboratory of Parasitology, Department of Disease Control, Graduate School of Infectious Diseases, Faculty of Veterinary Medicine, Hokkaido University, N 18 W 9, Kita-ku, Sapporo 060-0818, Japan; kobayashi.toshiya.1009@gmail.com (T.K.); chatanga@vetmed.hokudai.ac.jp (E.C.); kenkata@vetmed.hokudai.ac.jp (K.K.); nnonaka@vetmed.hokudai.ac.jp (N.N.); 2Department of Veterinary Pathobiology, Faculty of Veterinary Medicine, Lilongwe University of Agriculture and Natural Resources, Lilongwe P.O. Box 219, Malawi; 3Division of International Research Promotion, International Institute for Zoonosis Control, Hokkaido University, N 20 W10, Kita-ku, Sapporo 001-0020, Japan; yongjin_qiu@czc.hokudai.ac.jp; 4Department of Diseases Control, School of Veterinary Medicine, The University of Zambia, Lusaka P.O. Box 32379, Zambia; martin.simuunza@unza.zm (M.S.); nsaasa@gmail.com (N.S.); esikabala@yahoo.com (E.S.); atakada@czc.hokudai.ac.jp (A.T.); h-sawa@czc.hokudai.ac.jp (H.S.); 5Africa Centre of Excellence for Infectious Diseases of Humans and Animals, The University of Zambia, Lusaka P.O. Box 32379, Zambia; mudenda68@yahoo.com; 6Division of Global Epidemiology, International Institute for Zoonosis Control, Hokkaido University, N 20 W10, Kita-ku, Sapporo 001-0020, Japan; kajihara@czc.hokudai.ac.jp (M.K.); y.eto@frontier.hokudai.ac.jp (Y.E.); akinam@czc.hokudai.ac.jp (A.M.-K.); 7Department of ParaClinical Studies, School of Veterinary Medicine, The University of Zambia, Lusaka P.O. Box 32379, Zambia; 8Macha Research Trust, Choma P.O. Box 630166, Zambia; 9Division of Molecular Pathobiology, International Institute for Zoonosis Control, Hokkaido University, N 20 W10, Kita-ku, Sapporo 001-0020, Japan

**Keywords:** *Coxiella*-like endosymbionts, ticks, Zambia

## Abstract

Ticks are obligate ectoparasites as they require to feed on their host blood during some or all stages of their life cycle. In addition to the pathogens that ticks harbor and transmit to vertebrate hosts, they also harbor other seemingly nonpathogenic microorganisms including nutritional mutualistic symbionts. Tick nutritional mutualistic symbionts play important roles in the physiology of the host ticks as they are involved in tick reproduction and growth through the supply of B vitamins as well as in pathogen maintenance and propagation. *Coxiella*-like endosymbionts (CLEs) are the most widespread endosymbionts exclusively reported in ticks. Although CLEs have been investigated in ticks in other parts of the world, there is no report of their investigation in ticks in Zambia. To investigate the occurrence of CLEs, their maintenance, and association with host ticks in Zambia, 175 ticks belonging to six genera, namely *Amblyomma, Argas, Haemaphysalis, Hyalomma, Ornithodoros,* and *Rhipicephalus,* were screened for CLEs, followed by characterization of CLEs by multi-locus sequence typing of the five *Coxiella* housekeeping genes (*dnaK*, *groEL*, *rpoB*, 16S rRNA, and 23S rRNA). The results showed that 45.7% (n = 80) were positive for CLEs. The comparison of the tick 16S rDNA phylogenetic tree with that of the CLEs concatenated sequences showed that there was a strong correlation between the topology of the trees. The results suggest that most of the CLEs have evolved within tick species, supporting the vertical transmission phenomenon. However, the negative results for CLE in some ticks warrants further investigations of other endosymbionts that the ticks in Zambia may also harbor.

## 1. Introduction

Ticks are obligate ectoparasites as they require to feed on host blood during some or all stages of their life cycle [1]. Ticks harbor and transmit a variety of pathogens to their vertebrate hosts, including protozoa such as *Babesia* and *Theileria*, bacteria such as *Anaplasma*, *Borrelia, Ehrlichia*, and *Rickettsia*, and viruses such as Crimean-Congo haemorrhagic fever virus [2,3,4]. Tick-borne pathogens (TBPs) are one of the emerging public health concerns worldwide as they pose a threat to both humans and animals [5,6]. Thus, the need to investigate the microorganisms that ticks harbor is important in order to understand their epidemiology, transmission dynamics, and maintenance in the host ticks, which are prerequisites to conceiving effective tick control strategies.

In addition to pathogenic organisms that ticks harbor and transmit to both animals and humans, they also harbor a wide range of seemingly nonpathogenic microbes including symbionts [7,8,9]. Tick symbionts have attracted much attention among the scientists in the tick research community over the last decade and are now being investigated more than before due to their importance in the physiology of the ticks. Symbiotic bacteria are involved in tick reproduction and growth through the supply of vitamins such as biotin and folate [7,10,11] and are also increasingly known to be involved in pathogen maintenance and propagation in various vector arthropods [12,13]. Tick nutritional mutualistic endosymbionts are passed on to their offspring through vertical transmission [14,15]. Currently, three genera namely: *Coxiella*-like endosymbionts (CLEs), *Francisella*-like endosymbionts (FLEs), and *Midichloria* [8,9,16,17] have been reported exclusively in ticks. Initially, *Coxiella burnetii,* the causative agent of Q fever, was the only species under the genus *Coxiella*, but a nutritional mutualistic bacterium genetically related to *C. burnetii* was found among tick endosymbionts [18] and was termed CLE. The presence of CLEs has been reported from tick salivary glands, ovaries, and eggs [7,19,20,21]. Several studies have also reported the detection and genetic characterization of CLEs from a wide range of tick species [18,19,20,22,23,24,25,26]. 

In Zambia, a country located in southern Africa, 16 species of ticks belonging to two families (Argasidae and Ixodidae) and five genera (*Ornithodoros*, *Amblyomma*, *Haemaphysalis*, *Hyalomma* and *Rhipicephalus*) were initially reported, comprising of one species of *Ornithodoros*, three species of *Amblyomma*, one species of *Haemaphysalis*, two species of *Hyalomma*, and nine species of *Rhipicephalus* [27]. In recent years, other tick species have also been reported in Zambia, which include one species of *Argas*, one species of *Haemaphysalis*, one species of *Hyalomma*, one species of *Ornithodoros*, and eight species of *Rhipicephalus* [28], bringing the total to 28 species belonging to two families and six genera. Tick-borne diseases (TBDs) such as bovine theileriosis (or East Coast Fever) caused by *Theileria parva* infection, bovine babesiosis (or red water) caused by *Babesia bovis* infection, bovine anaplasmosis (or gall sickness) caused by *Anaplasma marginale* infection, and heartwater caused by *Ehrlichia ruminantium* infection are some of the obstacles to livestock development in Zambia [29,30,31,32]. Zoonotic TBPs such as *Anaplasma platys*, *Borrelia*-like organism, the causative agent of human borreliosis, and *Rickettsia africae* have also been reported in Zambia [33,34,35,36]. 

There has been no attempt to detect and genetically characterize CLEs in ticks in Zambia. Thus, the lack of genetic information of CLEs in ticks in Zambia warrants new research in this area. Elucidating the microbiological properties of CLEs and their role in ticks will lead to a better understanding of tick physiology and survival strategies in nature, leading to the establishment of novel tick control methods. This study aimed to investigate the presence of CLEs in ticks in Zambia and to understand the mode of transmission, maintenance, and the association of CLEs with their host ticks by genotyping.

## 2. Results

### 2.1. Morphological Tick Identification

A total of 175 ticks belonging to two families (Argasidae and Ixodidae) and six genera (*Amblyomma*, *Argas*, *Haemaphysalis*, *Hyalomma*, *Ornithodoros*, and *Rhipicephalus*) were identified morphologically under a steromicroscope using established keys [27]. At the species level, the following were identified (*Amblyomma pomposum*, *Amblyomma variegatum*, *Argas walkerae*, *Haemaphysalis aciculifer*, *Haemaphysalis elliptica*, *Hyalomma marginatum*, *Hyalomma marginatum rufipes*, *Hyalomma truncatum*, *Ornithodoros faini*, *Rhipicephalus appendiculatus*, *Rhipicephalus camicasi*, *Rhipicephalus decoloratus*, *Rhipicephalus evertsi evertsi*, *Rhipicephalus geigyi*, *Rhipicephalus lunulatus*, *Rhipicephalus microplus*, *Rhipicephalus muhsamae*, *Rhipicephalus sanguineus*, *Rhipicephalus simus*, *Rhipicephalus sulcatus*, *Rhipicephalus turanicus*, and unclassified *Rhipicephalus* spp.) for a total of 22 known and other unclassified *Rhipicephalus* spp.

### 2.2. Molecular Identification and Phylogenetic Analysis of Ticks Based on the Mitochondrial 16S rRNA Gene Sequences

Molecular tick species identification was conducted by amplifying the partial sequences of the mitochondrial 16S ribosomal RNA gene (rDNA) from at least one sample from each tick species. The results were in good agreement with the morphological identification although sequences from the same tick species were not always identical. Generally, our sequences clustered together with those sequences of the same tick species in the GenBank. *Amblyomma pomposum*, *Am. variegutum*, *Hae. aciculifer*, and *Hae. elliptica*, belonged to the same clade. Most *Rhipicephalus* species clustered according to their species except the members of the *R. sanguineus* sensu lato (s.l.); *R. camicasi*, *R. guilhoni*, *R. sanguineus*, *R. sulcatus*, and *R. turanicus* [37,38], which clustered into a single clade (Figure 1).

### 2.3. Screening of CLE

The nested polymerase chain reaction (PCR) assays targeting the heat shock protein gene (*groEL*) of CLEs showed that 80/175 (45.7%) ticks belonging to 20 species from five genera were positive (Table 1). *Amblyomma pomposum*, *Hae. aciculifer*, *R. camicasi*, *R. evertsi evertsi*, *R. guilhoni*, *R. muhsamae*, and *R. simus* had 100% detection rates of CLEs. For *Am. variegatum*, *Hae. elliptica*, *Hy. truncatum*, *O. faini*, *R. appendiculatus*, *R. decoloratus*, *R. geigyi*, *R. lunulatus*, *R. microplus*, *R. sanguineus*, *R. sulcatus*, *R. turanicus*, and *Rhipicephalus* spp. had varying infection rates between 9% and 67%. However, *Ar. walkerae*, *Hy. marginatum rufipes*, and *Hy. marginatum* were all negative for CLEs. 

### 2.4. Multi-Locus Sequence Typing of groEL, dnaK, rpoB, 16S rRNA, and 23S rRNA Genes of CLE

Based on the results of the screening for CLEs, we conducted multi-locus sequence typing (MLST) on CLE-positive samples by targeting four additional housekeeping genes: chaperone protein DnaK (*dnaK*), RNA polymerase beta-subunit **(***rpoB*), 16S rRNA, and 23S rRNA. When comparing the PCR success rates among four target genes, we observed that 23S rRNA and *rpoB* had higher success rates 98% (n = 78) and 93% (n = 74), respectively. In contrast, *dnaK* and 16S rRNA could only amplify 49 and 42 samples, representing success rates of 61% and 53%, respectively. The percentages of variable sites for the five genes were 38.9%, 37.6%, 50.8%, 13.2%, and 25.1% for *dnaK*, *groEL*, *rpoB*, 16S rRNA, and 23S rRNA, respectively.

We obtained 22 unique sequences of the *groEL* gene from 80 ticks belonging to 20 species from five genera; we also obtained 22 unique sequences of the *dnaK* gene from 49 ticks belonging to 15 species from three genera. In the *rpoB* gene 23 unique sequences were obtained from 74 ticks belonging to 19 species from four genera, while in the 16S rRNA gene we obtained 27 unique sequences from 42 ticks belonging to 15 species from three genera, and 26 unique sequences of the 23S rRNA gene from 78 ticks belonging to 18 species from three genera. 

The 22 unique sequences of the *groEL* gene were designated as alleles G1 to G22. The 22 unique sequences of the *dnaK* gene were designated as alleles D1 to D22. The 23 unique sequences of the *rpoB* gene were designated as alleles R1 to R23. The 27 unique sequences of the 16S rRNA gene were designated as alleles 16S-1 to 16S-27, and the 26 unique sequences of the 23S rRNA gene were designated as alleles 23S-1 to 23S-26 (Table 2). The assignment of the alleles was based on the number of sequences obtained per allele i.e., the highest number of identical alleles were assigned allele 1 and the numbering continued in descending order of number of sequences. Where single sequences were obtained, the alleles were assigned in alphabetical order of the tick species. 

Although each tick species had species specific unique alleles of CLEs, some exceptions were observed where the same allele of CLEs was conserved among different tick genera or species. For example, in the *dnaK* gene, allele D3 was shared by CLE from *Am. variegatum*, *R. appendiculatus*, *R. lunulatus*, and unclassified *Rhipicephalus* spp. and in the *groEL* gene, G2 was shared by CLE from *Am. pomposum*, *Am. variegatum*, and *Hy. truncatum* (Table 2).

The phylogenetic analysis based on the CLE alleles obtained for each gene were constructed for 16S rRNA and *groEL* genes in Figure 2 and Figure 3, respectively, and *dnaK*, 23S rRNA, and *rpoB* genes in Appendix A, respectively. The results have shown that CLE alleles from *Rhipicephalus* species generally clustered together while those from *Amblyomma* species clustered with *Haemaphysalis* species. Some alleles of CLEs from *Hae. elliptica*, *R. muhsamae* clustered together with the pathogenic *C. burnetii* in all five genes. However, one allele of CLEs from *O. faini* also clustered with *C. burnetii* in the *groEL* tree, which was the only gene in which CLE was amplified in the tick species. In the *groEL* gene tree, allele G1 was shared by *R. guilhoni*, *R. sanguineus*, and *R. sulcatus*, while allele G8 was shared by *Hae. elliptica*, *R. guilhoni*, and *R. muhsamae*. Furthermore, allele G2 was shared by *Am. variegatum*, *Am. pomposum*, and *Hy. truncatum*. Finally, allele G6 was shared by *Am. pomposum* and *Hae. aciculifer*. In the 16S rRNA gene, there were no CLE alleles that were shared by different tick species. In the *rpoB* gene, only allele R6 was shared by *R. geigyi* and *R. decoloratus*. In the *dnaK* gene, alleles D2, D7, D20, and D22 were shared by different tick species. In the 23S rRNA gene, allele 23S-4 was shared by *R. turanicus*, *R. camicasi*, and *R. guilhoni* while allele 23S-12 was shared by *Hae. elliptica* and *R. muhsamae*.

### 2.5. Comparison of Alleles of CLE and Their Phylogenetic Relationship with the Host Ticks

The samples that were successfully sequenced for all five loci were concatenated to produce a matrix of all the five genes. The concatenated sequences were used to construct a phylogenetic tree (Figure 4). The CLEs harbored by ticks of the genus *Rhipicephalus* generally clustered according to the species. However, the CLEs from *Am. pomposum* and *Hae. aciculifer* clustered together.

The comparison of the phylogenetic tree based on the concatenated five CLE genes with phylogenetic tree based on the mitochondrial 16S rRNA genes of ticks is shown in Figure 4. The phylogenetic divergence formed from CLEs of ticks was consistent with the phylogenetic divergence of their host ticks.

## 3. Discussion

The purpose of this study was to determine the prevalence of CLEs in ticks collected in Zambia and to analyze their relationship with host ticks by genotyping. The CLEs were detected in 80 ticks belonging to 20 species from five genera, including two species of *Amblyomma*, two species of *Haemaphysalis*, one species of *Hyalomma*, one species of *Ornithodoros*, and 13 species of *Rhipicephalus* and other unclassified *Rhipicephalus* spp. On the African continent, CLEs have been reported in a number of tick species, which include *Amblyomma cohaerens*, *Amblyomma gemma*, *Amblyomma lepidum*, *Amblyomma personatum*, *Amblyomma tholloni*, *Amb. variegatum*, *Haemaphysalis leachi*, *Haemaphysalis* sp., *Hy. truncatum*, *R. appendiculatus*, *Rhipicephalus carnivoralis*, *Rhipicephalus compositus*, *R. evertsi evertsi*, *Rhipicephalus maculatus*, *Rhipicephalus praetextatus*, *Rhipicephalus pravus*, *R. sanguineus* s.l., and *Rhipicephalus* sp. [39,40]. This is the first study that investigated CLE in *Am. pomposum*, *Hae. aciculifer*, and *O. faini*.

The *groEL* PCR assays were used for screening our samples because it is one of the genes that has been used extensively in the studies of CLEs in ticks [19,20,23]. The CLE positive detection rates were 98%, 93%, 61%, and 53% for *2*3S rRNA, *rpoB*, *dnaK*, and 16S rRNA genes, respectively, when tested using the 80 samples that were positive on *groEL* screening PCR. A prior study by Duron et al. [20] showed that in a sub-sample of 85 *Coxiella*-positive ticks, they were able to obtain multi-locus sequences in 84% (*n* = 71) of the genes 16S rRNA, 23S rRNA, *groEL*, and *dnaK*. Our findings have shown that 23S rRNA and *rpoB* PCR were more sensitive at amplifying CLE-positive samples when compared to *dnaK* and 16S rRNA PCR in our samples. This suggests that although all the genes targeted for amplification are *Coxiella* housekeeping genes, the PCR assays for some genes may not be robust enough and lead to false negatives during the amplification of CLEs in this study. Thus, for classification or typing of CLEs, there is a need to develop novel robust MLST primer sets by obtaining the genomic data of CLEs from all the clades of CLEs so that they encompass the genetic diversity of CLEs. It is also possible to use highly comprehensive methods such as bacterial species composition analysis by 16S rRNA gene amplicon analysis to detect diverging CLEs than the use of conventional PCR. This approach is supported by the findings of the study where the detection sensitivity of specific bacterial species harbored by ticks was compared between conventional PCR and 16S rDNA amplicon analysis and showed that the sensitivity was higher in the latter [3]. 

The number of alleles of CLEs obtained in this study were 22, 23, 23, 27, and 26 for *groEL*, *rpoB*, *dnaK*, 16S rRNA, and 23S rRNA genes, respectively. The *groEL* gene, despite having the highest number of samples successfully sequenced, had the least number of alleles, indicating that this gene is a good marker for screening CLEs and *C. burnetii*. In contrast, despite having only 42 samples successfully sequenced, 16S rDNA had the highest number of alleles at 27. Though 16S rDNA had the least percentage of variable sites at 13.2% compared to the other four genes examined, the high number of alleles obtained further supports the use of this genetic marker in characterizing CLEs. 

Although the tick species showed some species-specific alleles of CLEs, some alleles of CLEs were shared among different tick species. Similarly, the phylogenetic analysis showed that the clades of CLEs were made mainly based on tick genus. Clade A, which includes human pathogenic *C. burnetii* [20] comprised of *C. burnetii* reference strains and the sequences obtained from *Hae. elliptica* based on 16S rDNA-based tree *(*Figure 2*)*, while it included the sequences from *Hae. elliptica*, *O. faini*, *R. guilhoni*, and *R. muhsamae* in a tree based on the *groEL* gene (Figure 4). Clade B, where CLEs of *Haemaphysalis* ticks are present along with a presumably pathogenic *Coxiella* [41], was mainly composed of the sequences obtained from *Amblyomma* and *Haemaphysalis* in both 16S rDNA and *groE* gene-based trees (Figure 2 and Figure 3). Clade C, which includes CLEs from *R. turanicus* (CRt), a pathogen-derived endosymbiont along with strains causing opportunistic human skin infections [42,43,44,45,46], was comprised of the sequences obtained from *Rhipicephalus* in both 16S rDNA and *groEL* gene-based trees (Figure 2 and Figure 3). Similar clustering patterns were also observed in the trees based on other genes (Appendix A). The phylogenetic relationship of concatenated CLE genes was generally consistent with the phylogenetic tree of the tick’s 16S rDNA (Figure 4). The existence of a bias in which sequences of the CLEs possessed by different tick species were conserved within a particular tick species and the strong pattern of co-cladogenesis between ticks and CLEs may be due to vertical transmission of CLEs to the offspring of the ticks [47,48,49]. In the 16S rRNA gene, we did not observe any alleles that were shared by different tick species. This may be due to the least number of samples being successfully sequenced in this study as well as the low number of variable sites observed in the sequences obtained.

*Coxiella burnetii* is a Gram-negative intracellular pathogen that has evolved to invade and survive in vertebrate cells [20,50]. Previously, it was suggested that CLE underwent an evolutionary process to *C. burnetii* through genetic mutations and the acquisition of genes that define virulence from other pathogens [23]. However, recent comparative genomic studies supported the opposite view that CLEs have evolved from pathogenic *Coxiella* independently at multiple time points [51,52]. Nonetheless, it cannot be ruled out that the CLE obtained in this study can have the potential to adapt to vertebrate hosts like *C. burnetii*. This hypothesis is supported by previous studies, where sequences of CLE obtained from horse blood samples and ticks collected from the horses were identical [50]. Furthermore, an experimental study on the endosymbiont *Midichloria mitochondrii* in *Ixodes ricinus* ticks provided evidence of transmission of the endosymbionts to vertebrate hosts during blood feeding [53]. These findings also suggest that the influx of CLE from outside may occur by horizontal transmission through arthropod bites when feeding on host animals. In addition, the detection of symbionts in tick’s salivary glands may further support this phenomenon [53]. However, further studies are warranted to validate this hypothesis.

We conducted the phylogenetic analysis of ticks based on their mitochondrial 16S rDNA sequences (Figure 1). Similar to a previous report on ticks in Japan [54], we found that the clusters were divided by genus and species, except for members of the *R. sanguineus* s.l.: *R. camicasi*, *R. sanguineus*, *R. sulcatus*, *R. lunulatus*, and *R. muhsamae* which were mixed in the same clade, indicating that the amplified region of 16S rDNA is not suited to discriminate highly related species. Similarly, morphological identification of these ticks remains a big challenge among the tick research community, which is also supported by the report by Dantas-Torres et al. [37] that morphological identification of *R. turanicus* based on only spiracular plates does not correlate with molecular findings. Since most of these ticks resemble each other and due to the lack of type-material and no *bona fide* morphological description, this has resulted in them being referred to as *R. sanguineus* s.l. [37,55]. This may explain why there is a mix-up among these tick species. In order to understand the exact phylogenetic position of these tick species, detailed phylogenetic analysis using more genes such as complete mitochondrial genome sequencing is required in the future. For instance, the complete mitochondrial genome analysis by Liu et al. [56] provided the evidence that *R. sanguineus* has a number of other closely related species, which cluster together with it hence referred to as *R. sanguineus* complex (sensu lato). Furthermore, complete mitochondrial genome can also help to detect cryptic tick species [57]. 

The negative results for CLEs in *Ar. walkerae*, *Hy. marginatum rufipes*, and *Hy. marginatum* (Table 1) may be due to presence of other symbionts or divergent CLE species that could not be captured by the PCR primers used in this study. Other tick endosymbionts including *Rickettsiella*, *Midichloria*, *Lariskella*, *Francisella*, *Arsenophonus*, *Cardinium*, *Wolbachia*, *Rickettsia*, and *Spiroplasma* have also been reported in ticks [8,9,16,17]. In *Hy. marginatum,* the endosymbionts FLEs and *Midichloria* have been reported [9]. Furthermore, we also detected *Rickettsia* species in the *Ar. walkare* samples used in this study in our previous study [58]. Similarly, the pathogenic *C. burnetii* has been reported in Zambia in dogs and rodents [59] but not in ticks. Another study also reported the presence of FLEs in *Ornithodoros moubata* [60]. Thus, this study does not rule out the presence of other endosymbionts in these ticks. 

In this study, we clarified the genetic diversity of ticks and CLEs harbored by ticks collected in Zambia and evaluated the phylogenetic relationship between them. Our results showed a strong pattern of co-cladogenesis between ticks and CLEs, confirming the previous results that the vertical route of inheritance is the main one for these nutritional mutualistic symbionts. Thus, further studies are required to investigate the possibility of horizontal transmission of CLEs in particular and the tick endosymbionts in general. This will lead to a better understanding of the physiological characteristics and genetic phylogenetic relationships of ticks and CLEs, which may be applied to the development of tick control methods.

## 4. Materials and Methods

### 4.1. Ethical Consideration

Permission to sample ticks was obtained from the Department of Veterinary Services according to the Animal Health Act No. 27 of 2010 of the Laws of Zambia. Ticks were only sampled from those farmers who agreed to have their cattle and pastures sampled.

### 4.2. Tick Collection, Morphological Identification, and DNA Extraction

A total of 175 individuals ticks were collected from nine sampling sites in Zambia between January 2016 and December 2017 using the flannel flagging method on vegetation and picking up from the body surfaces of livestock (Figure 5). A total of 73 ticks were collected from the environment, while 71, 25, and six ticks were collected from cattle, dogs, and goats, respectively. We included only adult ticks that were apparently not engorged. The ticks were morphologically identified to the species level under a stereomicroscope using established keys [27]. Tick DNA was extracted using TRIzol Reagent (Thermo Fisher Scientific, Waltham, MA, USA) according to manufacturer’s recommendations. The concentration of the extracted DNA was measured using Qubit dsDNA BR Assay kit (Thermo Fisher Scientific). The extracted DNA was kept at −20 °C until required for use. 

### 4.3. Molecular Tick Identification

To complement the morphological identification of the ticks and for phylogenetic analysis, PCR amplification targeting tick 16S rDNA was conducted on the utmost three selected samples per tick species according to the previous study [61]. In brief, the PCR was performed using Tks Gflex Polymerase (Takara Bio Inc., Shiga, Japan) in a 10.00 µL reaction mixture containing 0.50 µL DNA template [1.0–10.0 ng], 5.00 µL of 2× Gflex Buffer, 0.20 µL of Tks Gflex Polymerase, 0.20 µL of each primer (10 µM), and 3.90 µL of molecular grade water. The PCR condition were initial denaturation at 94 °C for 1 min, followed by 40 cycles of denaturation at 98 °C for 10 s, annealing temperature for 15 s, extension at 68 °C for 1 min, and final extension at 68 °C for 5 min. The amplicons were electrophoresed in 1.5% agarose gel stained with Gel-Red (Biotium, Hayward, CA, USA) and visualized under UV light. 

### 4.4. Detection of CLE and Typing

All 175 tick samples were tested for the presence of CLEs using a nested PCR assay and sequencing of the *groEL* gene using *Coxiella*-specific primers as previously described [19,20]. All the primers used in this study, their annealing temperatures, and expected amplicon sizes are indicated in Table 3.

For the 1st *groEL* PCR, amplification was carried out in a total of 25.00 µL reaction mixture containing 1.00 µL of tick DNA template [2.0–20 ng], 12.50 µL of 2× Gflex Buffer, 0.50 µL of Tks Gflex Polymerase, 0.50 µL of each primer (10 µM), and 10.00 µL of distilled water. The reaction conditions as described above except that the annealing temperature was adjusted to 56 °C. The 2nd *groEL* PCR was performed in a 10.00 µL reaction mixture containing 0.50 µL of 1st PCR product as DNA template, 5.00 µL of 2× Gflex Buffer, 0.20 µL of Tks Gflex Polymerase, 0.20 µL of each primer (10 µM), and 3.90 µL of molecular grade water. The reaction conditions were the same as that of the 1st PCR. The amplicons were electrophoresed in 1.5% agarose gel stained with Gel-Red and visualized under UV light. 

The samples that were positive for CLEs were further subjected to MLST analysis using nested PCR assays targeting four other *Coxiella* genes: *dnaK*, *rpoB*, 16S rRNA, and 23S rRNA genes as described previously [20]. The PCRs were performed using Tks Gflex Polymerase. The reaction conditions were set as previously described with adjustments in annealing temperatures only. When the initial 1st or 2nd PCR failed, we did an alternative PCR to ensure that we successfully amplify and sequence as many samples as possible. For 16S rDNA of CLEs we amplified the 1421 bp fragment in two fragments of 939 bp and 627 bp for fragment 1 and fragment 2, respectively.

### 4.5. Sequencing

The PCR amplicons with single band were purified using ExoSAP-IT (Thermo Fisher Scientific, Santa Clara, CA, USA). When the amplicons in which multiple bands were identified, the band containing the target DNA was cut from the gel and purified using Nucleospin^®^ Gel & PCR clean-up (Macherey-Nagel., Neumann-Neander, Düren, Germany).

The amplicons of tick 16S rDNA PCR and 2nd PCR for each gene of CLE were sequenced in both directions using forward and reverse primers. The sequencing primers of CLE 16S rDNA were designed in this study. Sequencing reactions were performed in a 10.00 µL reaction mixture containing 1.00 µL of purified PCR product, 1.75 µL of 5× Sequencing Buffer (Applied Biosystems, Foster City, CA, USA), 0.50 µL of BigDye Terminator version 3.1 Cycle Sequencing Kit (Applied Biosystems), 0.32 µL of primer (10 µM), and 6.43 µL of distilled water. The sequencing products were purified using Agencourt CleanSeq (Beckman Coulter Inc., Brea, CA, USA) and sequenced on 3130xL Genetic Analyzer (Applied Biosystems).

### 4.6. Data Analysis

Sequences were analyzed using GENETYX version 9.1 (GENETYX Corporation, Tokyo, Japan), they were trimmed on both the 5’ and 3’ ends to remove the primer annealing sites as described previously [62]. The number and percentage of variable sites were calculated using DnaSP v6 [63]. The consensus CLE and tick 16S rDNA sequences obtained in this study were submitted to the DNA Data Bank of Japan (DDBJ) under accession numbers (*groEL* gene: LC634776–LC634852; *dnaK* gene: LC634853–LC634906; *rpoB* gene: LC634907–LC634982; 16S rDNA: LC635154–LC635193; and 23S rDNA: LC635194–LC635266, tick mitochondrial 16S rDNA: LC634544–LC634602). Then, phylogenetic trees were constructed in MEGA 7 [64] using the maximum likelihood method with the Kimura 2-parameter model. To test for confidence, the bootstrap values were calculated using 1000 replications. 

## 5. Conclusions

We confirmed that 80 ticks of 20 species in five genera harbored CLEs, but ticks belonging to three species in two genera were all negative for CLE. Each tick species showed predisposition to specific alleles of CLE with some exceptions. The phylogenetic relationship between tick species and their CLE was generally consistent. From this study, it was confirmed that many tick species in Zambia possess CLE. On the other hand, the existence of tick species that do not possess CLE was also confirmed, suggesting the existence of other symbiotic bacteria which may perform similar functions. Comparison of the phylogenetic relationship between ticks and CLE suggested that most of the CLE were inherited by vertical transmission, but there was also a possibility that CLE were introduced from outside by horizontal transmission. In future studies, clarification of the microbiological characteristics of CLE is expected to lead to a better understanding of the physiological characteristics of ticks.

## Figures and Tables

**Figure 1 pathogens-10-00779-f001:**
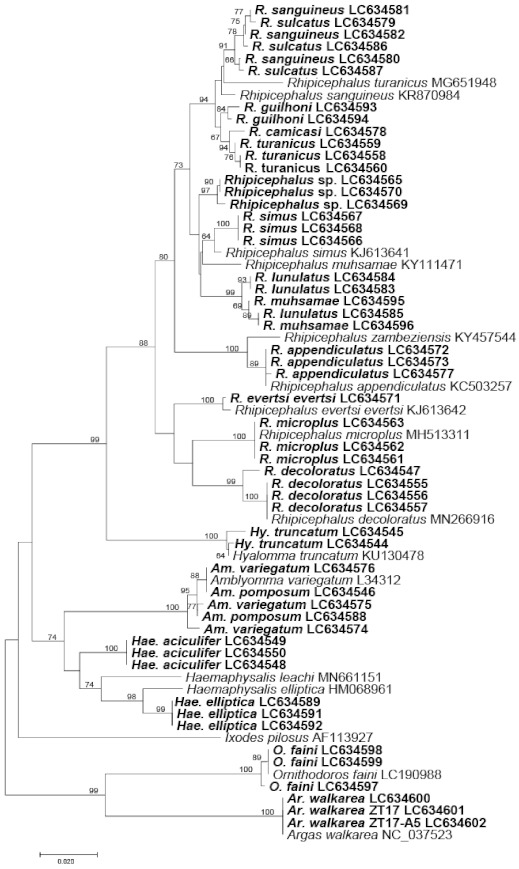
The phylogenetic tree based on the tick mitochondrial 16S rRNA gene partial sequences. The tree was constructed using MEGA7 based on the maximum likelihood method, using the Kimura 2-parameter model. All bootstrap values >60 from 1000 replications are shown on the interior branch nodes. The sequences obtained in this study are in bold. GenBank accession number is provided next to the tick species name.

**Figure 2 pathogens-10-00779-f002:**
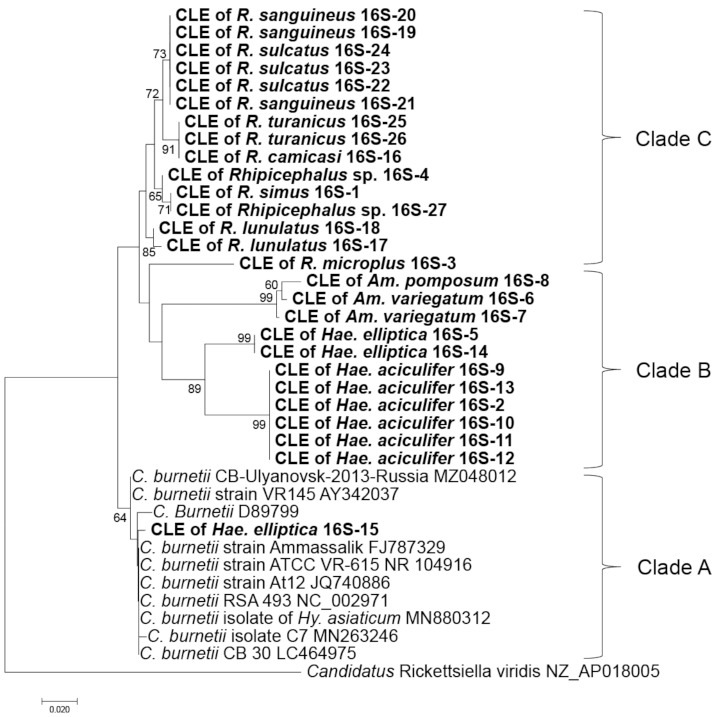
Phylogenetic tree based on the sequences of the CLE 16S rRNA gene. The phylogenetic tree was constructed using MEGA 7 based on the maximum likelihood method, using the Kimura 2-parameter model. All bootstrap values > 60 from 1000 replications are shown on the interior branch nodes. The sequences obtained in this study are in bold. Allele ID is provided next to the tick species name.

**Figure 3 pathogens-10-00779-f003:**
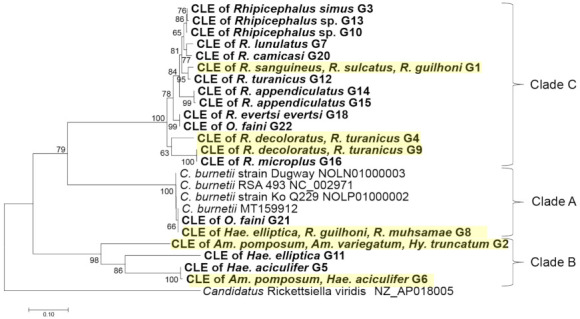
Phylogenetic tree based on the sequences of the CLE *groEL* gene. The phylogenetic tree was constructed using MEGA7 based on the maximum likelihood method, using the Kimura 2-parameter model. All bootstrap > 60 values from 1000 replications are shown on the interior branch nodes. The sequences obtained in this study are in bold. Alleles that were shared by different tick species have been highlighted in yellow. Allele ID is provided next to the tick species name.

**Figure 4 pathogens-10-00779-f004:**
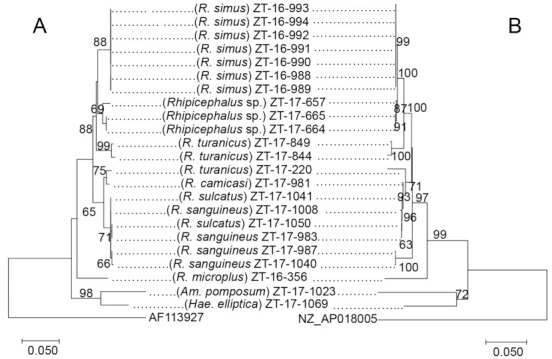
Comparison of phylogenetic trees based on the tick mitochondrial 16S rRNA gene partial sequences (**A**) and the CLEs concatenated sequences of five (*dnaK*, *groEL*, *rpoB*, 16S rRNA, and 23S rRNA) genes (**B**), the trees are rooted with *Ixodes pilosus* (AF113927) and *Candidatus* Rickettsiella viridis (NZ_AP018005), respectively. The trees were constructed using MEGA7 based on the maximum likelihood method, using the Kimura 2-parameter model. All bootstrap values >60 from 1000 replications are shown on the interior branch nodes. Sample ID is provided next to the tick species name.

**Figure 5 pathogens-10-00779-f005:**
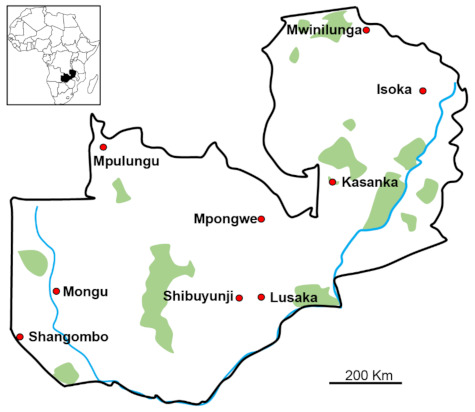
Map of Zambia showing the districts where tick samples were collected.

**Table 1 pathogens-10-00779-t001:** List of tick species, details on their origin, host species or habitat, number tested, the prevalence of CLEs, and year of collection.

Tick Species	Sampling Site	Host or Habitat	Number Tested (M; F)	Number Positive (M; F)	Positive Rate (%)	Year
*Amblyomma pomposum*	Mpulungu, Shangombo	dog, cattle	2 (1; 1)	2 (1; 1)	100	2016, 2017
*Amblyomma variegatum*	Mpulungu, Shangombo	cattle	13 (7; 6)	7 (3; 4)	54	2016, 2017
*Argas walkarae*	Isoka	vegetation	10 (NA)	0	0	2017
*Haemaphysalis aciculifer*	Kasanka	unknown	9 (5; 4)	9 (5; 4)	100	2017
*Haemaphysalis elliptica*	Mpulungu, Shibuyunji	vegetation, unknown	10 (6; 4)	4 (2; 2)	40	2017
*Hyalomma marginatum*	Mongu, Shangombo	cattle	6 (1; 5)	0	0	2017
*Hyalomma marginatum rufipes*	Shibuyunji	unknown	1 (1; 0)	0	0	2017
*Hyalomma truncatum*	Shangombo, Shibuyunji	cattle, vegetation	11 (4; 7)	1 (0; 1)	9	2016, 2017
*Ornithodoros faini*	Lusaka	cave	10 (4; 6)	2 (1; 1)	20	2017
*Rhipicephalus appendiculatus*	Mpongwe, Mpulungu, Shangombo	cattle, vegetation	10 (6; 4)	4 (2; 2)	40	2016, 2017
*Rhipicephalus camicasi*	Mpongwe	goat	1 (0; 1)	1 (0; 1)	100	2017
*Rhipicephalus decoloratus*	Mpongwe	cattle	10 (5; 5)	4 (2; 2)	40	2017
*Rhipicephalus evertsi evertsi*	Mpongwe	cattle	1(0; 1)	1 (0; 1)	100	2017
*Rhipicephalus geigyi*	Mongu	cattle	5 (0; 5)	1 (0; 1)	20	2017
*Rhipicephalus guilhoni*	Mpulungu	vegetation	2 (2; 0)	2 (2; 0)	100	2017
*Rhipicephalus lunulatus*	Isoka, Shangombo	dog, goat	12 (5; 7)	6 (2; 4)	50	2016, 2017
*Rhipicephalus microplus*	Mpongwe	cattle	9 (3; 6)	3 (1; 2)	33	2017
*Rhipicephalus muhsamae*	Mpulungu	vegetation	2 (2; 0)	2 (2; 0)	100	2017
*Rhipicephalus sanguineus*	Mpulungu, Shangombo	dog	13 (6; 7)	7 (2; 5)	54	2016, 2017
*Rhipicephalus simus*	Mwinilunga	unknown	8 (5; 3)	8 (5; 3)	100	2017
*Rhipicephalus sulcatus*	Mpulungu	cattle	6 (4; 2)	4 (2; 2)	67	2017
*Rhipicephalus turanicus*	Mpongwe	vegetation	10 (7; 3)	4 (2; 2)	40	2017
*Rhipicephalus* spp.	Isoka	cattle	14 (8; 6)	8 (3; 5)	57	2017
**Totals**			175 (82; 83)	80 (38; 42)	45.7	

M, male; F, female; NA, not available.

**Table 2 pathogens-10-00779-t002:** Multi-locus sequence typing of five CLE genes.

Tick Species	CLE Target Gene														
	*groEL*			*dnaK*			*rpoB*			16S rRNA			23S rRNA		
	No. Tested	No. +ve	Allele Type	No. Tested	No. +ve	Allele Type	No. Tested	No. +ve	Allele Type	No. Tested	No. +ve	Allele Type	No. Tested	No. +ve	Allele Type
*Amblyomma pomposum*	2	2	G2, G6	2	1	D20	2	2	R11, R12	2	1	16S-6	2	2	23S-3
*Amblyomma variegatum*	13	7	G2	7	1	D20	7	7	R8, R11, R12, R19	7	2	16S-7, 16S-8	7	7	23S-3, 23S-14
*Haemaphysalis aciculifer*	9	9	G5, G6	9	4	D20, D21, D22	9	9	R2, R18	9	9	16S-2, 16S-9, 16S-10, 16S-11, 16S-12	9	9	23S-5, 23S-6, 23S-15
*Haemaphysalis elliptica*	10	4	G8, G11	4	1	D2	4	4	R7, R14, R17	4	4	16S-5, 16S-13, 16S-14	4	4	23S-7, 23S-12
*Hyalomma truncatum*	11	1	G2	1	-	-	1	1	R16	1	-	-	1	1	23S-16
*Ornithodoros faini*	10	2	G21, G22	2	-	-	2	-	-	2	-	-	2	-	-
*Rhipicephalus appendiculatus*	13	4	G14, G15	4	4	D9, D18, D19	4	4	R21, R24	4	-	-	4	4	23S-13, 23S-17, 23S-18
*Rhipicephalus camicasi*	1	1	G20	1	1	D16	1	1	R22	1	1	16S-15	1	1	23S-4
*Rhipicephalus decoloratus*	10	4	G4, G9	4	4	D4, D10	4	4	R6	4	-	-	4	4	23S-19
*Rhipicephalus evertsi evertsi*	1	1	G18	1	-	-	1	1	R21	1	-	-	1	1	23S-20
*Rhipicephalus geigyi*	5	1	G17	1	1	D11	1	1	R6	1	-	-	1	1	23S-21
*Rhipicephalus guilhoni*	2	2	G1, G8	2	1	D7	2	2	R15	2	-	-	2	2	23S-4
*Rhipicephalus lunulatus*	12	6	G7	6	3	D5, D6	6	6	R4	6	2	16S-16, 16S-17	6	6	23S-9, 23S-22, 23S-23
*Rhipicephalus microplus*	10	4	G9, G16	4	3	D8	4	4	R10	4	2	16S-3	4	4	23S-8
*Rhipicephalus muhsamae*	2	2	G8	2	1	D22	2	2	R7	2	-	-	2	2	23S-12
*Rhipicephalus sanguineus*	13	7	G1, G19	7	7	D2, D13, D14,	7	7	R3, R20	7	3	16S-18, 16S-19, 16S-20	7	7	23S-1, 23S-24
*Rhipicephalus simus*	8	8	G3	8	8	D1	8	8	R1	8	7	16S-1	8	8	23S-2
*Rhipicephalus sulcatus*	6	4	G1	4	2	D2	4	4	R3	4	3	16S-21, 16S-22, 16S-23	4	4	23S-1, 23S-7
*Rhipicephalus turanicus*	10	4	G4, G12	4	4	D7, D12	4	4	R9	4	3	16S-24, 16S-25	4	4	23S-4, 23S-25
*Rhipicephalus* spp.	10	7	G10, G13	7	3	D3, D15 D17,	7	3	R5	7	5	16S-4, 16S-26, 16S-27	7	7	23S-10, 23S-11, 23S-26

No., number; +ve, positive.

**Table 3 pathogens-10-00779-t003:** List of primers used in this study.

Target Gene	Primer Name	Sequence 5’→ 3’	PCR Type	Tm (°C)	Fragment Size (bp)	Reference
Ticks 16S rDNA	mt-rrs 1	CTGCTCAATGATTTTTTAAATTGCTGTGG	Single	55	401–416	[61]
	mt-rrs 2	CCGGTCTGAACTCAGATCAAGTA				
*Coxiella groEL*	Cox-GrF1	TTTGAAAAYATGGGCGCKCAAATGGT	1st PCR	56	655	[19]
	Cox-GrR2	CGRTCRCCAAARCCAGGTGC				
	Cox-GrF2	GAAGTGGCTTCGCRTACWTCAGACG	2nd PCR	56	619	[20]
	Cox-GrR1	CCAAARCCAGGTGCTTTYAC				
*Coxiella dnaK*	Cox-dnaKF1	CGTCARGCRACGAARGATGCA	1st PCR	54	777	[20]
	Cox-dnaKR	CGTCATGAYKCCGCCYAAGG				
	Cox-dnaKF3	GGTACKTTYGATATTTCCATC	Alternative 1st PCR	54	636	[20]
	Cox-dnaKR	CGTCATGAYKCCGCCYAAGG				
	Cox-dnaKF2	GAAGTGGATGGCGARCAYCAATT	2nd PCR	54	603	[20]
	Cox-dnaKR	CGTCATGAYKCCGCCYAAGG				
	Cox-dnaKF3	GGTACKTTYGATATTTCCATC	Alternative 2nd PCR	54	512	[20]
	Cox-dnaKR3	CTTGAATAGCYGCACCAATAGC				
*Coxiella rpoB*	Cox-rpoBF2	GGGCGNCAYGGWAAYAAAGGSGT	1st PCR	56	607–610	[20]
	Cox-rpoBR1	CACCRAAHCGTTGACCRCCAAATTG				
	Cox-rpoBF3	TCGAAGAYATGCCYTATTTAGAAG	2nd PCR	56	539–542	[20]
	Cox-rpoBR3	AGCTTTMCCACCSARGGGTTGCTG				
*Coxiella* 16S rDNA	Cox-16SF1	CGTAGGAATCTACCTTRTAGWGG	1st PCR	52–56	1321–1429	[19,20]
	Cox-16SR2	GCCTACCCGCTTCTGGTACAATT				
	16S-07F	AGAGTTTGATYMTGGCTCAG	Alternative 1st PCR	52–56	1434–1542	[19,20]
	Cox-16SR2	GCCTACCCGCTTCTGGTACAATT				
	Cox-16SF1	CGTAGGAATCTACCTTRTAGWGG	2nd PCR (fragment 1)	52–56	719–826	[20]
	Cox-16SR1	ACTYYCCAACAGCTAGTTCTCA				
	16S-07F	AGAGTTTGATYMTGGCTCAG	Alternative 2nd PCR (fragment 1)	52–56	832–939	[20]
	Cox-16SR1	ACTYYCCAACAGCTAGTTCTCA				
	Cox-16SF2	TGAGAACTAGCTGTTGGRRAGT	2nd PCR (fragment 2)	52–56	624–627	[20]
	Cox-16SR2	GCCTACCCGCTTCTGGTACAATT				
	Cox16S_seq1	TCTACGCATTTCACCGCTAC	Sequencing			This study
	Cox16S_seq2	AGTCGGATGTGAAAGCCCTA				
	Cox16S_seq3	CCTGTCACTCGGTTCCCAAA				
	Cox16S_seq4	CTGACACTGAGGCCGCGAAAGC				
*Coxiella* 23S rDNA	Cox-23SF1	GCCTGCGAWAAGCTTCGGGGAG	1st PCR	56	694–1188	[20]
	Cox-23SR2	CTCCTAKCCACASCTCATCCCC				
	Cox-23SF2	GATCCGGAGATWTCYGAATGGGG	2nd PCR run	56	583–867	[20]
	Cox-23SR1	TCGYTCGGTTTCGGGTCKACTC				
	Cox-23SF1	GCCTGCGAWAAGCTTCGGGGAG	Alternative 2nd PCR	56	601–884	[20]
	Cox-23SR2	CTCCTAKCCACASCTCATCCCC				

## Data Availability

The CLE and tick sequences obtained in this study were submitted to the DNA Data Bank of Japan (DDBJ) under accession numbers (*groEL* gene: LC634776–LC634852; *dnaK* gene: LC634853–LC634906; *rpoB* gene: LC634907–LC634982; 16S rDNA: LC635154–LC635193; and 23S rDNA: LC635194–LC635266, tick mitochondrial 16S rDNA: LC634544–LC634602).

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
