# Peer review of "Molecular Detection and Genotyping of Coxiella-Like Endosymbionts in Ticks Collected from Animals and Vegetation in Zambia"

_pathogens, 2021, doi:10.3390/pathogens10060779_

Round 1
Reviewer 1 Report
In the work
The authors use molecular tools (PCR, sequencing) to detect and characterize with MLST genes the CLE of ticks found in Zambia
COMMENTS
- The view that burnetii evolved from CLE has been recently challenged. Convincing genomic evidence show that it was very likely the other way around (Nardi et al 2021, Brenner et al 2021). Please add the relevant citations in the introduction and modify the discussion accordingly.
- The word symbionts should be used more cautiously, as it can be many things (according to a broad definition, parasites are symbionts too). So I would generally define ticks symbiont as nutritional endosymbionts, or even better nutritional mutualistic symbionts, so the boundaries are clearer.
- While no studies have been performed on CLE in Zambia, others have investigated symbionts in African ticks, see for example Oundo et al 2020, Olivieri et al 2021. The authors should scan the relevant literature and provide comparisons where possible. For example evaluating whether presence of CLE has been previously reported on the species they investigate.
- CLE are classified in clades. Based on your sequences, can you tell what clades your CLEs belong to? Evaluate this aspect, add this information to the trees and discuss the results.
MINOR
The English could be improved. I include some examples:
Line 35: (CLE) are the most widespread endosymbionts
Line 70-72: Rephrase, also considering the comments above
Figure 1. Explain what the codes are in the figure legends. I suppose ZT means this study while accessions are from pubmed.
Line 201: Matrix
Line 232: 16S metagenomics would allow to detect diverging CLE, but not to type them. Genomes of all clades are necessary to re-design a novel MLST primers set, one that encompasses the genetic diversity of CLE.
Line 246: What is the % of variable sites? This is a good way to measure if 16S is more variable than other genes.
Line 249: Improve the construction of this sentence please.
Line 252: What do you mean? Not clear.
Line 277: This is clearly due to 16S not being variable enough to discriminate between highly similar species (XX% variable sites between the aforementioned species of the complex).
Line 289: This tick is complex. What do you mean?
Line 292: I would say the presence of other symbionts is the most likely explanation, followed by the presence of diverging CLE your primers do not detect. Absence of symbionts is very unlikely.
Line 293: Francisella
Line 302: Rephrase. Something like ‘our results show a strong pattern of co-cladogenesis between ticks and CLE, confirming also for the analyzed species, previous results indicating the vertical route of inheritance as the main one for these symbionts.
Author Response
COMMENTS
Reviewer 1: The view that C. burnetii evolved from CLE has been recently challenged. Convincing genomic evidence show that it was very likely the other way around (Nardi et al 2021, Brenner et al 2021). Please add the relevant citations in the introduction and modify the discussion accordingly.
Author's Response: We agree with the comment, and the sentence has been revised in line 259-262 as follows: “Previously, it was suggested that CLE underwent an evolutionary process to C. burnetii through genetic mutations and the acquisition of genes that define virulence from other pathogens [23]. However, recent comparative genomic studies supported the opposite view that CLEs have evolved from pathogenic Coxiella independently at multiple time points [51, 52].
Reviewer 1: The word symbionts should be used more cautiously, as it can be many things (according to a broad definition, parasites are symbionts too). So I would generally define ticks symbiont as nutritional endosymbionts, or even better nutritional mutualistic symbionts, so the boundaries are clearer.
Author's Response: We agree with the suggestion made in the comment and some changes have been made accordingly but we have maintained the term Coxiella-like endosymbionts (CLE) as it is widely used.
Reviewer 1: While no studies have been performed on CLE in Zambia, others have investigated symbionts in African ticks, see for example Oundo et al 2020, Olivieri et al 2021. The authors should scan the relevant literature and provide comparisons where possible. For example evaluating whether presence of CLE has been previously reported on the species they investigate.
Author's Response: We agree with the comment and information on the investigation of CLE in other African countries has been included in lines 225-232 as follows: “On the African continent CLEs have been reported in a number of tick species which include Amblyomma cohaerens, Amblyomma gemma, Amblyomma lepidum, Amblyomma personatum, Amblyomma tholloni, Amb. variegatum, Haemaphysalis leachi, Haemaphysalis sp., Hy. truncatum, R. appendiculatus, Rhipicephalus carnivoralis, Rhipicephalus compositus, R. evertsi evertsi, Rhipicephalus maculatus, Rhipicephalus praetextatus, Rhipicephalus pravus, R. sanguineus s.l. and Rhipicephalus sp. [39, 40].
Reviewer 1: CLE are classified in clades. Based on your sequences, can you tell what clades your CLEs belong to? Evaluate this aspect, add this information to the trees and discuss the results.
Author's Response: We agree with the comment and we have assigned our sequences to three clades (A, B, and C) as shown in Figures 2 and 3 and Supplementary Figures 1, 2, and 3.
We also included the explanation in the Results as follows:
“Clade A, which includes human pathogenic C. burnetii [20] comprised of C. burnetii reference strains and the sequences obtained from Hae. elliptica based on 16S rDNA-based tree (Figure 2), while it included the sequences from Hae. elliptica, O. faini, R. guilhoni, and R. muhsamae in a tree based on groEL gene (Figure 4). Clade B, where CLEs of Haema-physalis ticks are present along with a presumably pathogenic Coxiella [41], was mainly composed of the sequences obtained from Amblyomma and Haemaphysalis in both 16S rDNA and groE gene-based trees (Figure 2 and Figure 3). Clade C, which includes CLE from R. turanicus (CRt) a pathogen-derived endosymbiont along with strains causing opportunistic human skin infections [42- 46], comprised of the sequences obtained from Rhipicephalus in both 16S rDNA and groE gene-based trees (Figure 2 and Figure 3). Similar clustering patterns were also observed in the trees based on other genes (Figures S1, S2, and S3).” (lines 262-273)
MINOR
Reviewer 1: Line 35: (CLE) are the most widespread endosymbionts
Author's Response: We agree with the suggestion and the sentence has been changed as follows: “Coxiella-like endosymbionts (CLE) are the most widespread endosymbionts exclusively reported in ticks.”
Reviewer 1: Line 70-72: Rephrase, also considering the comments above
Author's Response: We agree with the comment and the sentence has been rephrased as follows: “Initially, Coxiella burnetii, the causative agent of Q fever, was the only species under the genus Coxiella, but a nutritional mutualistic bacterium genetically related to C. burnetii was found among tick endosymbionts [18] and was termed CLE.”
Reviewer 1: Figure 1: Explain what the codes are in the figure legends. I suppose ZT means this study while accessions are from PubMed.
Author's Response: The codes have been explained in the legend of the figure as follows: “GenBank accession no. or sample ID is provided next to the tick species name.”
Reviewer 1: Line 201: Matrix
Author's Response: The spelling has been corrected from “matric” to “matrix”.
Reviewer 1: Line 232: 16S metagenomics would allow to detect diverging CLE, but not to type them. Genomes of all clades are necessary to re-design a novel MLST primers set, one that encompasses the genetic diversity of CLE.
Author's Response: We agree with the comment and in the above stated sentence we indicated that 16S metogenomics is more sensitive at detecting not typing bacterial species. Therefore, we changed the paragraph as follows: “This suggests that although all the genes targeted for amplification are Coxiella house-keeping genes the PCR assays for some genes may not be robust enough and lead to false negatives during the amplification of CLE in this study. Thus, for classification or typing of CLEs, there is a need to develop novel robust MLST primer sets by obtaining the genomic data of CLEs from all the clades of CLEs so that they encompass the genetic diversity of CLEs. It is also possible to use highly comprehensive methods such as bac-terial species composition analysis by 16S rRNA gene amplicon analysis to detect di-verging CLE than the use of 16S rDNA conventional PCR.” (lines 241-248)
Reviewer 1: Line 246: What is the % of variable sites? This is a good way to measure if 16S is more variable than other genes.
Author's Response: We agree with the comment and the percentage of variable sites for the five genes were 38.9%, 37.6%, 50.8%, 13.2%, and 25.1% for dnaK, groEL, rpoB, 16S rRNA, and 23S rRNA respectively. We provided these values in the Results (lines 150-152).
Reviewer 1: Line 249: Improve the construction of this sentence please.
Author's Response: We agree with the comment and we have revised the sentence as follows: “The groEL gene despite having the highest number of samples successfully sequenced had the least number of alleles, indicating that this gene is a good marker for screening CLE and C. burnetii.” (lines 253-255)
Reviewer 1: Line 252: What do you mean? Not clear.
Author's Response: We rephrased the sentences as follows:
“In contrast, despite having only 42 samples successfully sequenced, 16S rDNA had the highest number of alleles at 27. Though 16S rDNA had the least percentage of variable sites at 13.2% compared to the other four genes examined, the high number of alleles obtained further supports the use of this genetic marker in characterizing CLE.” (lines 255-259)
Reviewer 1: Line 277: This is clearly due to 16S not being variable enough to discriminate between highly similar species (5% variable sites between the aforementioned species of the complex).
Author's Response: We agree with the comment, the sentence has been revised as follows: “Similar to a previous report on ticks in Japan [54], we found that the clusters were divided by genus and species, except for members of the R. sanguineus sensu lato: R. camicasi, R. sanguineus, R. sulcatus, R. lunulatus, and R. muhsamae which were mixed in the same clade, indicating that amplified region of 16S rDNA is not suited to discriminate highly related species.” (lines 299-303).
Reviewer 1: Line 289: This tick is complex. What do you mean?
Author's Response: We agree with the comment, the word complex in this context means that Rhipicephalus sanguineus sensu lato is composed of a number of closely related tick species. The sentence as rephrased as follow:
“For instance, the complete mitochondrial genome analysis by Liu et al. [56] provided the evidence that R. sanguineus has a number of other closely related species which cluster together with it hence referred to as R. sanguineus complex (sensu lato)."
Reviewer 1: Line 292: I would say the presence of other symbionts is the most likely explanation, followed by the presence of diverging CLE your primers do not detect. Absence of symbionts is very unlikely.
Author's Response: We agree with the comment and the sentence has been revised to read as follows: “The negative results for CLE in Ar. walkerae, Hy. marginatum rufipes, and Hy. marginatum (Table 1) may be due to presence of other symbionts or divergent CLE species that could not be captured by the PCR primers used in this study”.
Reviewer 1: Line 293: Francisella
Author's Response: The spelling has been corrected from “Franscisella” to “Francisella”.
Reviewer 1: Line 302: Rephrase. Something like ‘our results show a strong pattern of co-cladogenesis between ticks and CLE, confirming also for the analyzed species, previous results indicating the vertical route of inheritance as the main one for these symbionts.
Author's Response: We agree with the comment and the sentence has been rephrased as suggested as follows: “Our results showed a strong pattern of co-cladogenesis between ticks and CLEs, confirming the previous results that the vertical route of inheritance is the main one for these nutritional mutualistic.” (lines 328-331)
Reviewer 2 Report
The authors investigated the presence of Coxiella-like endosymbionts (CLE) in native tick species of Zambia. They sampled ticks from the environment and animals. Tick species was determined based on morphology and 16S rRNA gene sequencing. Results were in good agreement.
Screening for CLE was done using groEL as target gene. Additionally four genes, dnaK, rpoB, 16S rRNA and 23S rRNA encoding genes were sequenced and phylogenetic trees calculated. This is an appropriate method to distinguish between C. burnetii and CLE. PCR success varies, and phylogenetic analysis was done with 22 sequences of groEL, 22 sequences from dnaK, 23 sequences of rpoB, 27 of 16S rRNA and 26 of 23S rRNA encoding genes.
Almost half of the ticks of various species carried CLE. For some tick species, all of the samples were positive. Phylogenetic analyses of CLE were done in comparison to C. burnetii Nine Mile only. The authors demonstrated that some CLE clustered together in accordance with their tick species host. This is a very interesting finding and should be investigated further. Despite vertical transmission, there might be other factors involved.
Major Comments:
The authors should include more than one C. burnetii reference strain in their phylogenetic analyses. They should also include CLE-specific gene sequences from different tick species available from gene bank for comparison. This would provide a better differentiation of CLE from C. burnetii.
Line 229: The authors conclude from low amplification success of some genes (dnaK, 16S rRNA), that these have mutations at the primer-binding site. Without sequencing data, this statement cannot be drawn just on amplification failure. There are several factors affecting PCR success such as DNA quality and concentration as well as robustness of the assay.
Line 241: The authors state that groEL is highly conserved and a good screening target for CLE. In the phylogenetic tree C. burnetii Nine Mile clusters together with CLE –GroEL-8. With a PCR targeting groEL also samples containing C. burnetii will be positive. In my understanding this is a tool for screening CLE and C. burnetii and further discrimination is needed.
Line 244: Do you mean high number of PCR assay cited in the literature?
Minor comments:
The author could provide a distribution map for tick collection sites.
The authors could provide information on how many ticks were collected from the environment and how many from animals and which animals.
Line 171: dnaK must be written in italics
Author Response
Review Report Form 2
The authors investigated the presence of Coxiella-like endosymbionts (CLE) in native tick species of Zambia. They sampled ticks from the environment and animals. Tick species was determined based on morphology and 16S rRNA gene sequencing. Results were in good agreement.
Screening for CLE was done using groEL as target gene. Additionally four genes, dnaK, rpoB, 16S rRNA, and 23S rRNA encoding genes were sequenced and phylogenetic trees calculated. This is an appropriate method to distinguish between C. burnetii and CLE. PCR success varies, and phylogenetic analysis was done with 22 sequences of groEL, 22 sequences from dnaK, 23 sequences of rpoB, 27 of 16S rRNA and 26 of 23S rRNA encoding genes.
Almost half of the ticks of various species carried CLE. For some tick species, all of the samples were positive. Phylogenetic analyses of CLE were done in comparison to C. burnetii Nine Mile only. The authors demonstrated that some CLE clustered together in accordance with their tick species host. This is a very interesting finding and should be investigated further. Despite vertical transmission, there might be other factors involved.
Major Comments:
Reviewer 2: The authors should include more than one C. burnetii reference strain in their phylogenetic analyses. They should also include CLE-specific gene sequences from different tick species available from gene bank for comparison. This would provide a better differentiation of CLE from C. burnetii.
Author's Response: We agree with the comment and we have included more C. burnetii reference strains in the phylogenetic trees in Figures 2 and 3 and supplementary Figures 1, 2, and 3.
Reviewer 2: Line 229: The authors conclude from low amplification success of some genes (dnaK, 16S rRNA), that these have mutations at the primer-binding site. Without sequencing data, this statement cannot be drawn just on amplification failure. There are several factors affecting PCR success such as DNA quality and concentration as well as robustness of the assay.
Author's Response: We agree with the observation made in the comment. However, since the same DNA was successfully amplified for tick mitochondrion 16S rRNA PCR assay and other genes of Coxiella specific PCR assays, it is our considered view that DNA quality and concentration may not have a major effect on PCR success rates but the robustness of these PCR assays may be a major contributory factor. This has been included in the discussion in lines 227-229 as follows: “This suggests that although all the genes targeted for amplification are Coxiella house-keeping genes the PCR assays for some genes may not be robust enough and lead to false negatives during the amplification of CLE in this study. Thus, for classification or typing of CLEs, there is a need to develop novel robust MLST primer sets by obtaining the genomic data of CLEs from all the clades of CLEs so that they encompass the genetic diversity of CLEs.” (lines 241-246)
Reviewer 2: Line 241: The authors state that groEL is highly conserved and a good screening target for CLE. In the phylogenetic tree C. burnetii Nine Mile clusters together with CLE –GroEL-8. With a PCR targeting groEL also samples containing C. burnetii will be positive. In my understanding this is a tool for screening CLE and C. burnetii and further discrimination is needed.
Author's Response: We agree with the comment and in this study we used other Coxiella housekeeping genes to further characterize our isolates. The sentence has been revised to read as follows: “The groEL gene despite having the highest number of samples successfully sequenced had the least number of alleles, indicating that this gene is a good marker for screening CLE and C. burnetii.” (lines 253-255).
Reviewer 2: Line 244: Do you mean high number of PCR assay cited in the literature?
Author's Response: This sentence has been revised as follows: “. In contrast, despite having only 42 samples successfully sequenced, 16S rDNA had the highest number of alleles at 27. Though 16S rDNA had the least percentage of variable sites at 13.2% compared to the other four genes examined, the high number of alleles obtained further supports the use of this genetic marker in characterizing CLE.” (lines 255-259)
Minor Comments:
Reviewer 2: The author could provide a distribution map for tick collection sites.
Author's Response: We agree with the comment and the map has been provided as Figure 5.
Reviewer 2: The authors could provide information on how many ticks were collected from the environment and how many from animals and which animals.
Author's Response: We agree with the comment and the information on number of ticks collected from the environment, animals and animal species has been provided in line 312-313 as follows: “A total of 73 ticks were collected from the environment, while 71, 25, and 6 ticks were collected from cattle, dogs, and goats, respectively. We included only adult ticks that were apparently not engorged.” (lines 345-347)
Reviewer 2: Line 171: dnaK must be written in italics
Author's Response: We agree with the comment and the word “dnaK” has been italicized.
Reviewer 3 Report
The manuscript ID: pathogens-1261488 entitled “Molecular detection and genotyping of Coxiella-like endosymbionts in ticks collected from animals and vegetation in Zambia” by Toshiya Kobayashi is focused to investigate the presence of Coxiella bacteria in ticks in Zambia and to understand the mode of transmission, maintenance, and the association with ticks by genotyping.
The manuscript is well written. Some minor revisions are suggested by the referee.
Lines 135-137: “According to the authors knowledge, this is the first report of the detection of CLE in Am. pomposum, Hae. aciculifer, and O. faini as there is no report of CLE investigation in these tick species”.
Some of the analysed specimens were collected from animals (i.e. ticks of A. pomposum and O. faini species). Which was the engorgement level of these ticks? Which was their life stage? I think that this point is important in order to affirm that these tick species really harbor CLE. In fact, PCRs positivity for the detection of CLE in ticks could be due to the blood meal instead of the effective positivity of the arthropod. Better explain this point.
Lane 145: change “amomg” with “among”
Lanes 322 336, 362: please, indicate a range of DNA concentration
Lane 328-329: move these two lines in the subsequent paragraphs (at line 333).
Line 364: indicate the concentration of the primers used in sequencing analysis
Line 374: close the round bracket
Author Response
Review Report Form 3
Comments and Suggestions for Authors
The manuscript ID: pathogens-1261488 entitled “Molecular detection and genotyping of Coxiella-like endosymbionts in ticks collected from animals and vegetation in Zambia” by Toshiya Kobayashi is focused to investigate the presence of Coxiella bacteria in ticks in Zambia and to understand the mode of transmission, maintenance, and the association with ticks by genotyping.
The manuscript is well written. Some minor revisions are suggested by the referee.
Reviewer 3: Lines 135-137: “According to the authors knowledge, this is the first report of the detection of CLE in Am. pomposum, Hae. aciculifer, and O. faini as there is no report of CLE investigation in these tick species."
Author's Response: The sentence has been revised as follows: “To our knowledge, this is the first study that has investigated the presence of CLE in Am. pomposum, Hae. aciculifer, and O. faini.”
Reviewer 3: Some of the analysed specimens were collected from animals (i.e. ticks of A. pomposum and O. faini species). Which was the engorgement level of these ticks? Which was their life stage? I think that this point is important in order to affirm that these tick species really harbor CLE. In fact, PCRs positivity for the detection of CLE in ticks could be due to the blood meal instead of the effective positivity of the arthropod. Better explain this point.
Author's Response: We agree with the comment, since it is difficult to rule out the possibility of the detected CLE to be from the animals that the ticks feed on. Only those ticks that were not engorged were used in this study, all ticks were adult both male and female were included in this study as provided in Table 1. We added the following sentence in the Materials and methods: “A total of 73 ticks were collected from the environment, while 71, 25, and 6 ticks were collected from cattle, dogs, and goats, respectively. We included only adult ticks that were apparently not engorged.” (lines 345-347).
Reviewer 3: Line 145: change “amomg” with “among”
Author's Response: The spelling has been corrected to “among”
Reviewer 3: Lines 322 336, 362: please, indicate a range of DNA concentration.
Author's Response: We agree with the comment and the ranges of DNA concentrations is provided as 2.0 -20.0 ng.
Reviewer 3: Line 328-329: Move these two lines in the subsequent paragraphs (at line 333).
Author's Response: We agree with the comment and the two lines have been moved to the subsequent paragraph.
Reviewer 3: Line 364: indicate the concentration of the primers used in sequencing analysis
Author's Response: We agree with the comment and the primer concentration of 10 µM has been provided.
Reviewer 3: Line 374: close the round bracket
Author's Response: We agree with the comment and the bracket has been closed.